# Safety and Efficacy of a Large-Bore Biliary Metallic Stent for Malignant Biliary Obstruction

**DOI:** 10.3390/jcm11113092

**Published:** 2022-05-30

**Authors:** Min Young Do, Sung Ill Jang, Jae Hee Cho, Yonsoo Kim, In-Jung Kim, Kwang-Hun Lee, Seung-Moon Joo, Dong Ki Lee

**Affiliations:** 1Department of Internal Medicine, Gangnam Severance Hospital, Yonsei University College of Medicine, Seoul 06273, Korea; dmy24@yuhs.ac (M.Y.D.); aerojsi@yuhs.ac (S.I.J.); jhcho9328@yuhs.ac (J.H.C.); minpolaris@gmail.com (Y.K.); sginjung@naver.com (I.-J.K.); 2Department of Radiology, Gangnam Severance Hospital, Yonsei University College of Medicine, Seoul 06273, Korea; doctorlkh@yuhs.ac (K.-H.L.); huchi79@yuhs.ac (S.-M.J.)

**Keywords:** malignant biliary obstruction, fully covered, self-expandable metallic stent (FCSEMS), large-bore, dumbbell-shaped FCSEMS

## Abstract

Self-expandable metallic stents (SEMSs) are typically inserted in patients with unresectable malignant biliary obstruction. However, SEMSs are susceptible to occlusion. To overcome this issue, we developed a large-bore, dumbbell-shaped, fully covered SEMS (FCSEMS-L) and compared its efficacy and safety with those of a conventional FCSEMS (FCSEMS-C) in patients with malignant biliary obstruction. Methods: Patients with unresectable distal malignant biliary obstruction were retrospectively enrolled between January 2011 and February 2021. All patients underwent endoscopic insertion of FCSEMSs. Recurrent biliary obstruction (RBO), patient survival time, complications, and prognosis were analyzed. Results: RBO occurred in 31 patients (35.6%) who received an FCSEMS-L, and in 34 (45.9%) who received an FCSEMS-C. Stent occlusion occurred in 19 patients (21.8%) who received an FCSEMS-L, and in 22 (29.7%) who received an FCSEMS-C. Stent migration occurred in 12 patients (13.8%) with an FCSEMS-L and 12 (16.2%) with an FCSEMS-C. The median time to RBO (TRBO) was 301 days with an FCSEMS-L and 203 days with an FCSEMS-C. The median survival time was 479 days with an FCSEMS-L and 523 days with an FCSEMS-C. The TRBO and patient survival time did not significantly differ between the two groups. Conclusions: There were no significant differences in efficacy and complication rates between the fully covered large bore SEMSs and conventional fully covered SEMSs.

## 1. Introduction

Biliary obstruction is a common problem in patients with cholangiocarcinoma, pancreatic adenocarcinoma, and other malignancies [1]. Malignant biliary obstruction can cause debilitating symptoms such as pruritus and malaise [2], as well as an unfavorable prognosis and increased morbidity and mortality [3]. Therefore, various approaches for biliary decompression—including endoscopic, percutaneous, and surgical methods—have been used in patients with biliary obstruction [2,4]. Endoscopic placement of biliary stents was first performed in the 1980s [5]. Subsequently, there was a rapid shift from surgery towards endoscopic retrograde cholangiopancreatography (ERCP) stent placement because of its effectiveness and safety [6]. In patients with unresectable malignant biliary obstruction, endoscopic biliary stent placement is particularly suitable for relieving obstructive symptoms and improving morbidity and mortality [7,8].

ERCP-guided stent placement through the papilla is a standard palliative treatment for malignant distal biliary obstruction. Plastic stents (PSs) have reduced patency because of sludge formation and their narrow lumen [9]. Self-expandable metallic stents (SEMSs) were developed to overcome the limitations of PSs and improve patency [10,11,12]. Fully covered self-expandable metallic stents (FCSEMSs) are frequently used because of their increased biliary patency [11,12,13,14]. However, FCSEMSs are susceptible to occlusion by tumor ingrowth, overgrowth, and sludge formation, and have a high migration rate [15,16].

Large-diameter stents reduce stent occlusion by preventing sludge accumulation [17,18,19,20]. Uncovered SEMSs (UCSEMSs) with a body diameter of 14 mm body can be used to safely and effectively treat malignant biliary obstruction [20]. Also, the use of a 12-mm covered self-expandable end bare metal stent (CSEEMS) was associated with a lower risk of recurrent biliary obstruction (RBO) than a 10-mm FCSEMS in patients with unresectable malignant biliary obstruction [21].

A modified FCSEMS with a body diameter of 12 mm and head diameter of 16 mm has been developed. To prevent stent migration, the modified FCSEMS was designed with dumbbell-shaped heads at both ends. We evaluated the efficacy and safety of this modified large-bore stent (FCSEMS-L) compared with a conventional FCSEMS (FCSEMS-C).

## 2. Materials and Methods

### 2.1. Patients

This was a retrospective and comparative study involving patients with malignant biliary duct obstructions treated with biliary metallic stents between January 2011 and February 2021. The inclusion criteria were as follows: adults aged 19–90 years; patients with unresectable distal (≥2 cm distal to the biliary hilum) malignant biliary obstruction; estimated survival time of at least three months; Eastern Cooperative Oncology Group Performance Status (ECOG-PS) grade < 2; and willingness to provide written informed consent and comply with the follow-up requirements. Patients who met any of the following criteria were excluded: malignant hilar and/or intrahepatic duct stricture; ECOG-PS grade ≥ 3; previous surgical biliary drainage or SEMS; contraindication for ERCP; or pregnant or currently breastfeeding. The endpoints of this study were clinical outcomes, complications, and prognosis.

Outpatient follow-up was performed for all enrolled patients. Clinical symptoms and laboratory tests, including liver function tests (levels of total bilirubin, aspartate aminotransferase, alanine aminotransferase, alkaline phosphatase, and gamma-glutamyl transpeptidase), were performed before and immediately, three and seven days after stent insertion (and monthly thereafter). The location and length of the biliary stricture were confirmed through computed tomography (CT) and cholangiogram. CT was performed before and six months after stent insertion. When stent occlusion was clinically suspected during follow-up, CT and abdominal ultrasonography were performed.

### 2.2. Stents

The FCSEMS-L (Hilzo Biliary Stent^®^; BCM Ltd., Seoul, Korea) has a 12-mm-diameter body and 16-mm-diameter dumbbell-shaped heads (Figure 1A). Unique dumbbell-shape heads are designed to reduce migrations in the biliary tract. The FCSEMS-L membrane was polytetrafluoroethylene (PTFE). The diameter of the delivery system of undeployed FCSEMS-L is 8.5-Fr. The FCSEMS-C (Hanaro Biliary Stent^®^; M.I. Tech, Seoul, Korea) has a 10-mm-diameter head and 10-mm-diameter body (Figure 1B). The FCSEMS-C membrane was silicone. The diameter of the delivery system of undeployed FCSEMS-C is 8.5-Fr. An FCSEMS-L or FCSEMS-C was inserted endoscopically by two experienced investigators (D.K.L, S.I.J) (Figure 2). During ERCP, endoscopic sphincterotomy was performed before stent insertion in all patients. The stent length (4, 5, 6, 7, or 8 cm) was at the discretion of the investigators.

### 2.3. Definitions

Definitions were based on the Tokyo criteria (2014) for transpapillary biliary stenting [22]. Technical success was defined as the successful deployment of the FCSEMS in the intended location with sufficient coverage of the stricture. Clinical success was defined as a 50% decrease in, or normalization of, the bilirubin level within 14 days of stent placement. RBO was defined by the composite endpoint of either occlusion or migration. Stent occlusion was determined based on ERCP findings. Tumor ingrowth was defined as the direct growth of a tumor through the stent mesh, and tumor overgrowth as the growth of a tumor into the end of the stent. Stent migration referred to proximal or distal displacement of the stent from the initial insertion site. The time to RBO and patient survival were defined as the time from stent placement to the recurrence of biliary obstruction and death of the patient, respectively. Complications were categorized as early (within 30 days) or late (31 days or later).

### 2.4. Statistical Analysis

Data are presented as means ± standard deviations, numbers (percentage), or medians (range). Differences between the two groups were compared by independent-samples *t*-tests for continuous variables and chi-squared tests for categorical variables. The Kaplan-Meier method was used to evaluate the time to RBO and patient survival time. The log-rank test was used to calculate *p*-values in the Kaplan-Meier analysis. All *p*-values were two-sided. Values of *p* < 0.05 were considered indicative of statistical significance. A statistical analysis was performed using SPSS software (version 25.0; IBM Corp., Armonk, NY, USA).

## 3. Results

### 3.1. Patient Characteristics

Of the 161 patients, 87 underwent the insertion of an FCSEMS-L and 74 underwent the insertion of an FCSEMS-C (Table 1). The baseline characteristics of the two groups were similar in terms of age (FCSEMS-L, 67.1 ± 10.7 years; FCSEMS-C, 69.0 ± 13.2 years; *p* = 0.341) and gender (FCSEMS-L, 45 males; FCSEMS-C, 33 males; *p* = 0.367). The underlying disease and TNM stage did not significantly differ between the groups. Malignant tumor was pathologically confirmed in all patients.

The mean length of the biliary stricture was 24.3 ± 9.9 mm, and the mean maximum diameter of the proximal bile duct was 14.9 ± 5.0 mm in the FCSEMS-L group. The mean length of the biliary stricture was 26.4 ± 11.2 mm, and the mean maximum diameter of the proximal bile duct was 15.0 ± 4.9 mm in the FCSEMS-C group. There were no significant differences between the two groups in those parameters. The follow-up duration (FCSEMS-L, 261.9 ± 220.1 days; FCSEMS-C, 225.7 ± 233.8 days; *p* = 0.313), rate of chemotherapy, and the serum bilirubin level did not differ significantly between the two groups.

### 3.2. Clinical Outcomes

The technical and clinical success rates were 100% in both groups. In the FCSEMS-L group, 40-, 50-, 60-, 70-, and 80-mm-long stents were placed in 2, 9, 36, 24, and 16 patients, respectively. In the FCSEMS-C group, 50-, 60-, 70-, and 80-mm-long stents were placed in 7, 28, 24, and 15 patients, respectively; there were no significant differences between the two groups (Table 2). The incidence of RBO did not significantly differ between the two groups (35.6% [31 patients] in the FCSEMS-L group and 45.9% [34 patients] in the FCSEMS-C group, *p* = 0.184). The stent occlusion and migration rates did not differ significantly between the two groups (21.8% in the FCSEMS-L group and 29.7% in the FCSEMS-C group, *p* = 0.252; and 13.8% in the FCSEMS-L group and 16.2% in the FCSEMS-C group, *p* = 0.667). The median time to RBO (TRBO) was 301 days (range: 3–666 days) in the FCSEMS-L group and 203 days (range: 2–563 days) in the FCSEMS-C group. The non-RBO rates at three, six, and 12 months were 59.8%, 28.7% and 12.6% in the FCSEMS-L group and 51.4%, 21.6%, and 4.1% in the FCSEMS C group, respectively. The TRBO was not significantly different between the two groups (log-rank test *p* = 0.104) (Figure 3A). The median survival time was 479 days (range: 30–956 days) in the FCSEMS-L group and 523 days (range: 20–1256 days) in the FCSEMS C group. The patient survival time did not significantly differ between the two groups (log-rank test *p* = 0.955) (Figure 3B).

In the FCSEMS-L group, early and late complications occurred in three (3.4%) patients each. In the FCSEMS-C group, early and late complications occurred in seven (9.5%) and four (5.4%) patients, respectively. There was no significant difference in the early or late complication rate between the two groups. Complications—including pancreatitis, cholangitis, and cholecystitis—were acceptably resolved by conservative management in both groups.

### 3.3. Reinterventions

In the FCSEMS-L group, reintervention was attempted in 26 (29.9%) patients (because of stent occlusion in 15 and stent migration in 11). In patients with stent migration, there was the migrated stent in eight patients, whereas none was found in three patients. The stents were removed under fluoroscopic guidance with a duodenoscope using endoscopic snares or grasping forceps. All stents were removed successfully and there were no technical difficulties. Stent occlusion was treated by insertion of a second FCSEMS (*n* = 15) or a PS (*n* = 4). Stent migration was treated by insertion of a second FCSEMS (*n* = 9) or a PS (*n* = 2). In the FCSEMS-C group, reintervention was attempted in 27 (36.5%) patients (because of stent occlusion in 17 and stent migration in 10). In patients with stent migration, there was the migrated stent in four patients, whereas there was none found in six patients. Stent occlusion was treated by insertion of a second FCSEMS (*n* = 14) or a PS (*n* = 3). Stent migration was treated by insertion of a second FCSEMS (*n* = 9) or a PS (*n* = 1).

## 4. Discussion

Stent occlusion is the main problem associated with endoprostheses and necessitates additional interventions. Therefore, biliary stents with novel designs are currently under development [9,23]. Large-diameter SEMS may be more resistant to sludge accumulation. A large-bore (12-mm-diameter) FCSEMS has been reported to be safe and effective for managing malignant distal obstructions [16]. Among 38 patients with unresectable malignant distal biliary obstruction who underwent endoscopic insertion of a modified FCSEMS, the median time to RBO was 184 days and the median survival time was 241 days. Stent occlusion occurred as a result of sludge formation and tumor in growth in two (5%) patients. In a study of the efficacy of a 12-mm-diameter CSEEMS used in 99 patients with malignant distal biliary obstruction to prevent RBO, the median TRBO was significantly longer than in patients treated using a 10-mm FCSEMS (232 vs. 139.5 days; log-rank test, *p* = 0.001) [21]. The occlusion rate was significantly lower in the 10-mm CSEEMS group than in the 10-mm FCSEMS group (9.1% vs. 33.3%; *p* = 0.009). However, although it was a comparative study, a 12-mm CSEEMS is a large-bore partially covered stent, whereas a 10-mm FCSEMS is a conventional fully covered stent. Additionally, the longer time to RBO in the 12-mm CSEEMS group may have been associated with the use of a new type of chemotherapy [21]. In this study, the incidence of stent occlusion did not significantly differ between the two groups. Furthermore, the number of cases of stent occlusion caused by sludge and food impaction was similar between the groups. Duodenal-biliary reflux, which includes food material, can lead to stent occlusion due to biliary sludge formation and food impaction. Biofilms form via the adherence of proteins and bacteria to a stent, resulting in sludge formation [24,25]. The pressure gradient between the bile duct and duodenum decreases as the diameter of the SEMS increases, causing reflux of the duodenal contents [24,25,26,27]. The dumbbell shape of the duodenal end (i.e., the part of the FCSEMS that protrudes into the duodenal lumen) of the 16-mm-diameter FCSEMS-L induced reflux of the duodenal contents, promoting stent occlusion.

UCSEMSs may prevent migration but are easily occluded and difficult to remove because of ingrowth or overgrowth of mucosal tissue. Although covered SEMSs (CSEMSs) were developed to decrease occlusion, they are prone to migration [28]. A recent meta-analysis of 14 trials showed no significant difference in primary stent patency or dysfunction between CSEMSs and UCSEMSs during the follow-up period, until primary stent dysfunction or patient death. However, the meta-analysis concluded that a CSEMS is a better choice for patients with malignant biliary obstruction because of its removability [29]. A modified FCSEMS, with flared flanges at both ends to anchor the stent, was developed to prevent migration [30,31]. Modified CSEMSs with dog-bone-shaped flange ends and FCSEMSs with dumbbell-shaped ends were developed to reduce the risk of stent migration for patients with malignant esophageal cancer and benign pancreatic duct stricture, respectively [32,33]. In this study, the FCSEMS-L had a 16-mm-diameter dumbbell-shaped head, which we expected to hamper migration. However, there were no significant differences between the two groups in stent migration rate (12 [13.8%] in the FCSEMS-L group and 12 [16.2%] in the FCSEMS-C group, *p* = 0.667). Therefore, the specially designed stent ends did not prevent stent migration. This finding could have several causes. First, fiber-rich foods are easily trapped in stents with a dumbbell-shaped duodenal end, thus increasing the traction force toward the duodenum and promoting stent migration. Second, covered metallic stents have intrinsic limitations irrespective of any modification.

Complication rates are similar for conventional 10-mm diameter CSEMSs and large-bore CSEMSs [16,20,21]. In a prospective multicenter study involving 38 patients with unresectable malignant distal biliary obstruction, a 12-mm-diameter FCSEMS was inserted [16]. The median maximum diameter of the proximal bile duct was 13 mm (range: 8–30 mm). The bile duct diameters in the two patients were 8 and 9 mm. They complained of abdominal pain after FCSEMS insertion, which was improved by pain medications. In a retrospective study involving 38 patients with unresectable distal biliary obstruction treated using 14-mm-diameter uncovered metallic stents, the median maximum diameter of the proximal bile duct was 13.5 mm (range: 7–20 mm) [20]. Despite the large diameter of the stent, the complication rate was similar to prior studies. Pancreatitis and cholecystitis did not occur in patients in whom a 12-mm CSEEMS was employed [21]. In this study, the incidence of complications in the large bore metallic stent group was acceptable and similar to that of the 10-mm conventional FCSEMS group. In addition, severe complications related to the small diameter of the CBD were not reported. Patients with a large-diameter proximal bile duct had been dilated sufficiently due to their unresectable distal malignant biliary structure. Also, no complications occurred due to overdilation, even in small-diameter proximal bile ducts.

This study had several limitations. First, it was a single center, retrospective, nonrandomized trial. Further randomized controlled trials with adequate sample sizes comparing FCSEMS-L and FCSEMS-C are needed. Second, the membrane material of the FCSEMSs and stent end design were different. However, the tumor ingrowth and overgrowth did not differ significantly between the two groups. Third, because the choice of stent type and diameter was at the discretion of the endoscopist, selection bias may have been present. This study also had several strengths. To our knowledge, no study has evaluated the efficacy of an FCSEMS with a body diameter of 12 mm, head diameter of 16 mm, and dumbbell-shaped ends in patients with unresectable malignant biliary obstruction. This is the first study to show the efficacy of an FCSEMS-L. In a recent study, TRBO for an 8 mm diameter FCSEMS was equivalent to that for a 10-mm-diameter FCSEMS [34]. Therefore, stent usage should not be limited only by the inner stent diameter. Larger studies considering stent diameter and design are needed.

## 5. Conclusions

There were no significant differences in efficacy and complication rates between the fully covered large bore SEMSs and conventional fully covered SEMSs. Further investigation is required to evaluate the optimal stent diameter and to prevent stent migration. Additionally, well-designed studies with longer follow-up periods are needed to confirm the efficacy of FCSEMS-L for unresectable malignant biliary obstruction.

## Figures and Tables

**Figure 1 jcm-11-03092-f001:**
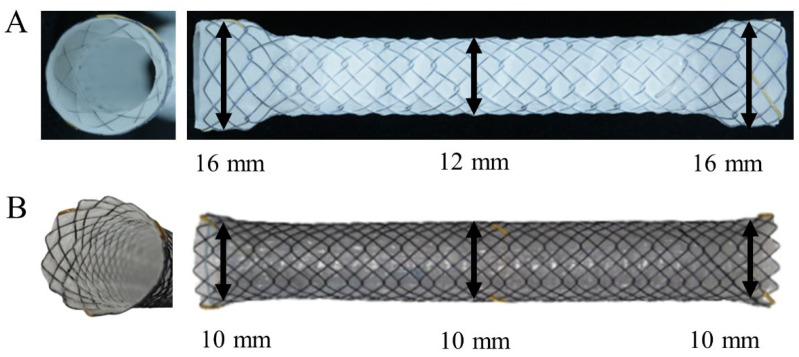
Structure of FCSEMSs. (**A**) A FCSEMS-L (Hilzo Biliary Stent^®^; BCM Ltd., Seoul, Korea) has a 12-mm body diameter and 16-mm head diameter, with dumbbell-shaped ends. (**B**) A FCSEMS-C (Hanaro Biliary Stent^®^; M.I. Tech, Seoul, Korea) has a 10-mm-diameter head and body.

**Figure 2 jcm-11-03092-f002:**
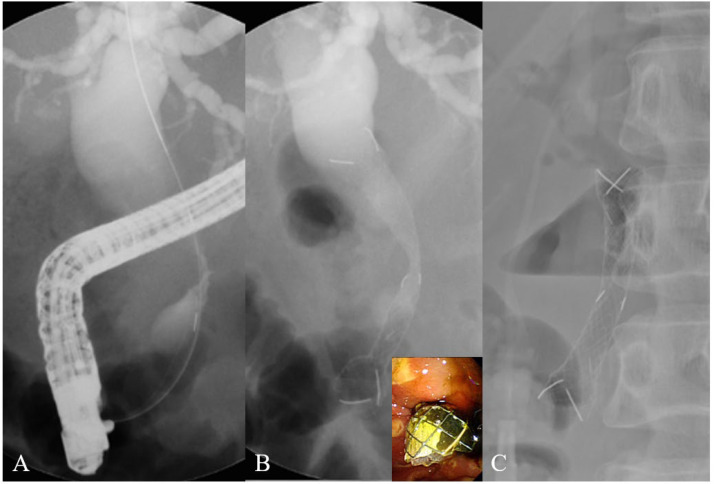
Insertion of a large-bore fully covered SEMS (FCSEMS-L). (**A**) The distal bile duct was obstructed by pancreatic cancer ingrowth, and the proximal bile duct was dilated. (**B**) The FCSEMS L was endoscopically inserted into the obstructed site. (**C**) The FCSEMS-L was fully deployed, and the bile duct was decompressed using an “air-biliarygram” three days after stent insertion.

**Figure 3 jcm-11-03092-f003:**
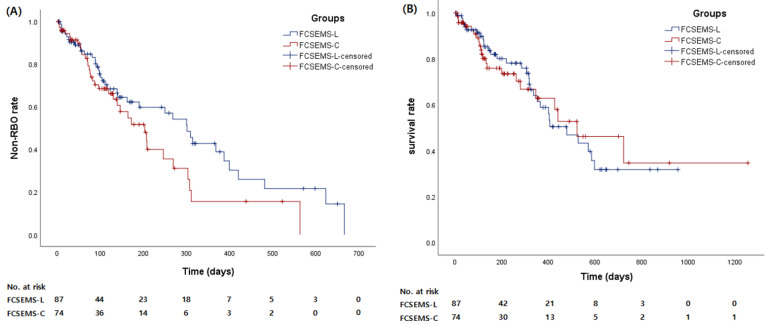
Time to recurrent biliary obstruction (RBO) and patient survival time in the FCSEMS-L and FCSEMS-C groups. (**A**) Time to RBO. (**B**) Patient survival time.

**Table 1 jcm-11-03092-t001:** Basic patient characteristics.

	FCSEMS-L (*n* = 87)	FCSEMS-C (*n* = 74)
Age, mean ± SD, years	67.1 ± 10.7	69.0 ± 13.2
Male: Female	45:42	33:41
Diagnosis, *n* (%)		
Pancreas cancer	56 (64.4)	39 (52.7)
CBD cancer	15 (17.2)	16 (21.6)
GB cancer	8 (9.2)	11 (14.9)
Ampulla of vater cancer	2 (2.3)	5 (6.8)
Metastatic disease	6 (6.9)	3 (4.1)
TNM Stage, *n* (%)		
III	33 (37.9)	35 (47.3)
IV	54 (62.1)	39 (52.7)
Duodenal stricture, *n* (%)	2 (2.3)	1 (1.4)
Length of biliary stricture, mean ± SD, mm	24.3 ± 9.9	26.4 ± 11.2
Maximum diameter of proximal bile duct, mean ± SD, mm	14.9 ± 5.0	15.0 ± 4.9
Follow-up period, mean ± SD, day	261.9 ± 220.1	225.7 ± 233.8
Chemotherapy, *n* (%)	68 (78.2)	49 (66.2)
Serum bilirubin level, mean ± SD, mg/dL		
Baseline	6.7 ± 5.5	7.0 ± 6.0
1 Day after stent insertion	4.6 ± 4.1	4.8 ± 4.3
4 Weeks after stent insertion	1.2 ± 0.7	1.4 ± 1.2

FCSEMS-L, large-bore fully covered self-expandable metallic stent; FCSEMS-C, conventional fully covered self-expandable metallic stent; SD, standard deviation; CBD, Common bile duct; GB, gallbladder; TNM, Tumor Nodes Metastases.

**Table 2 jcm-11-03092-t002:** Clinical outcomes of FCSEMS-L and FCSEMS-C.

Variables	FCSEMS-L(*n* = 87)	FCSEMS-C(*n* = 74)	*p* Value
Length of stent, *n* (%)			0.690
4 cm	2 (2.3)	0	
5 cm	9 (10.3)	7 (9.5)	
6 cm	36 (41.4)	28 (37.8)	
7 cm	24 (27.6)	24 (32.4)	
8 cm	16 (18.4)	15 (20.3)	
RBO, *n* (%)	31 (35.6)	34 (45.9)	0.184
Stent occlusion, *n* (%)	19 (21.8)	22 (29.7)	0.252
Sludge impaction	18	19	
Food impaction	10	7	
Tumor ingrowth	2	1	
Tumor overgrowth	1	4	
Stent migration, *n* (%)	12 (13.8)	12 (16.2)	0.667
Proximal	0	0	
Distal	12	12	
Early complication, *n* (%)	3 (3.4)	7 (9.5)	0.237
Pancreatitis	1	1	
Non-occlusion cholangitis	2	6	
Late complication, *n* (%)	3 (3.4)	4 (5.4)	0.808
Non-occlusion cholangitis	2	3	
Cholecystitis	1	1	

FCSEMS-L, large-bore fully covered self-expandable metallic stent; FCSEMS-C, conventional fully covered self-expandable metallic stent; RBO, recurrent biliary obstruction.

## Data Availability

The data that support the findings of this study are available from the corresponding author upon reasonable request.

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
