# Peer review of "Safety and Efficacy of a Large-Bore Biliary Metallic Stent for Malignant Biliary Obstruction"

_jcm, 2022, doi:10.3390/jcm11113092_

Round 1

Reviewer 1 Report

The phrase "biliary obstructions" appears in the abstract and in the main body text, and should be replaced with "biliary obstruction" which is grammatically correct. 

2.3 Definitions: You state using the 2014 Tokyo Criteria for the definitions used in this study.  The Tokyo Criteria define RBO as stent occlusion or symptomatic migration but your stated definition is just 'stent obstruction' ; please amend. 

4. Discussion:

Please spell out in full at the first use of the acronym CSEEMS.

The sentences referencing #32 (a study on modified esophageal CSEMS) and #33 (modified CSEMS for pancreatic ductal stricture) are not relevant to this paper on the utility of covered biliary SEMS, and should be removed (lines 218-229). The results of esophageal and pancreatic duct stent studies cannot be extrapolated to imply that the same results will apply to biliary stents. If desired, these papers may be referenced with regard to the type of modification made to reduce migration risk, and this can be done in a single sentence. 

Limitations: it should be stated that the stents that were compared differed in covering material, stent end design and diameter. 

Reviewer 2 Report

Introduction

Page 1, line 39: change “however” for “subsequently” 

Material and Methods: it would be very useful to indicate the diameter of the delivery system of undeployed stents. Both the large bore and the conventional one.

Results.

3.3 Reinterventions (page 7, line 172)

“stent removal was attempted in 26 (29.9%) patients (because of stent occlusion in 15 and stent migration in 11)”

Please indicate if migrated stents were not found at the second ERCP because migration was downwards and probably, they were expelled unnoticed in the faeces.

It does not appear appropriate to say that stent removal was attempted in cases of distal migration, If the stent did not migrate upwards (inside the bile duct).

The same reasoning must be said for both kind of stents

Discussion

Page 7, line 194

“The occlusion rate was significantly lower in  the 1-mm CSEEMS group than in the 10-mm FCSEMS”

I suppose the 1-mm is a mistake

Page 8, line 261

“In recent study” perhaps it should say: “in a recent study”

Line 267

“large bore SEMSs and conventional SEMSs”à fully covered large bore SEMSs and conventional fully covered SEMSs.
